# The Effects of Alcohol Drinking on Oral Microbiota in the Chinese Population

**DOI:** 10.3390/ijerph19095729

**Published:** 2022-05-08

**Authors:** Ying Liao, Xia-Ting Tong, Yi-Jing Jia, Qiao-Yun Liu, Yan-Xia Wu, Wen-Qiong Xue, Yong-Qiao He, Tong-Min Wang, Xiao-Hui Zheng, Mei-Qi Zheng, Wei-Hua Jia

**Affiliations:** 1State Key Laboratory of Oncology in South China, Collaborative Innovation Center for Cancer Medicine, Guangdong Key Laboratory of Nasopharyngeal Carcinoma Diagnosis and Therapy, Sun Yat-sen University Cancer Center, Guangzhou 510060, China; liaoying@sysucc.org.cn (Y.L.); tongxt@mail2.sysu.edu.cn (X.-T.T.); jiayj3@mail2.sysu.edu.cn (Y.-J.J.); liuqy77@mail2.sysu.edu.cn (Q.-Y.L.); wuyx23@mail2.sysu.edu.cn (Y.-X.W.); xuewq@sysucc.org.cn (W.-Q.X.); heyq@sysucc.org.cn (Y.-Q.H.); wangtm@sysucc.org.cn (T.-M.W.); zhengxh@sysucc.org.cn (X.-H.Z.); zhengmq@sysucc.org.cn (M.-Q.Z.); 2School of Public Health, Sun Yat-sen University, Guangzhou 510062, China

**Keywords:** oral microbiota, alcohol drinking, 16S rRNA gene sequencing, China

## Abstract

The dysbiosis of oral microbiota is linked to numerous diseases and is associated with personal lifestyles, such as alcohol drinking. However, there is inadequate data to study the effect of alcohol drinking on oral microbiota from the Chinese population. Here, we profiled the oral microbiota of 150 healthy subjects in the Chinese population by 16S rRNA gene sequencing. The results showed that drinkers had significantly higher alpha diversity than non-drinkers. A significant difference in overall microbiota composition was observed between non-drinkers and drinkers. Additionally, using DESeq analysis, we found genus *Prevotella* and *Moryella*, and species *Prevotella melaninogenica* and *Prevotella tannerae* were significantly enriched in drinkers; meanwhile, the genus *Lautropia*, *Haemophilus* and *Porphyromonas*, and species *Haemophilus parainfluenzae* were significantly depleted in drinkers. PICRUSt analysis showed that significantly different genera were mainly related to metabolism pathways. The oxygen-independent pathways, including galactose, fructose and mannose metabolism pathways, were enriched in drinkers and positively associated with genera enriched in drinkers; while the pyruvate metabolism pathway, an aerobic metabolism pathway, was decreased in drinkers and negatively associated with genera enriched in drinkers. Our results suggested that alcohol drinking may affect health by altering oral microbial composition and potentially affecting microbial functional pathways. These findings may have implications for better understanding the potential role those oral bacteria play in alcohol-related diseases.

## 1. Introduction

The oral cavity harbors complex bacterial communities. The effects of the oral micro-ecosystems extend far away from the oral cavity. Previous studies have identified several oral microbiome participates in oral and systemic diseases, such as bacteremia, cardiovascular disease, diabetes, and cancer [1,2,3,4,5]. Owing to the role of the oral microbiome in human health, identifying the factors that influence the microbiome is undoubtedly necessary to understand its roles in their associated diseases.

Alcohol consumption is considered a leading risk factor for disease burden, with 5.3% of all global deaths attributable to alcohol [6]. Health outcomes associated with alcohol consumption include breast cancer, ischemic heart disease, diabetes, and tuberculosis [7]. The oral cavity is the first part of the body that comes into contact with alcohol. Continuous alcohol consumption will alter the oral ecosystem including reducing saliva production, decreasing oral pH, and disrupting tooth enamel, resulting in cavities and periodontal inflammation [8,9]. The main constituent of alcohol is ethanol. Ethanol has direct cytotoxic effects on bacteria and is used as a substrate for bacterial metabolism [10,11]. Thus, alcohol might be one of the potential determinants in shaping the oral microbiota communities.

Previous studies have observed the association between alcohol drinking and oral microbiota. In a large population-based study of 1044 American adults, Fan et al. observed that the amount and type of alcohol consumption both have an impact on oral microbiota [12]. In addition, they found oral pathogens enriched in subjects with higher alcohol consumption. Signoretto et al. conducted a study on 75 Swiss adults and observed red wine can influence the microbiota at both the supragingival and the subgingival levels [13]. Thomas et al. observed that alcohol drinking will significantly reduce bacterial richness in the oral biofilm and alter the abundance of some bacteria in 22 American subjects [14]. Mary Rodríguez-Rabassa et al. have studied the salivary bacterial composition and revealed a higher abundance of *Prevotella* in the alcohol group [15]. Alcoholic drinks in China have some differences from those in the West, with common types including baijiu, beer, and rice wine. The microbiome is affected by humans living under different climatic conditions [16]. Exploring the association in diverse populations is necessary to identify whether the association of oral microbiome with alcohol consumption is universalized across different populations.

To improve our understanding of the influence of alcohol drinking on the oral microbiota, we conducted an oral microbial study of 150 subjects by 16S rRNA gene sequencing. This study may provide evidence of the relationship between alcohol drinking and oral microbiota, helping us to understand how alcohol exposure adversely affects human health.

## 2. Materials and Methods

### 2.1. Study Population and Saliva Sample Collection

Participants were drawn from the Chinese Environment, EBV, and Cancer Study (CEEC) project, which is described in detail in the previous study [17]. For this project, 1223 adults were recruited between 1 October 2015 and 1 August 2016 in Guangdong province, with a mean age (±SD) of 46.74 ± 11.16 years. Individuals were eligible if they were aged 20–80 years and were free from any history of cancer, immunological diseases, or acute diseases. At the enrollment step, saliva samples and baseline questionnaires were collected. Informed consent was signed by every subject before the interview, and the Human Ethics Committee of Sun Yat-sen University Cancer Center reviewed and approved the proposal for the study (the approval number: GZR2013-008).

All 150 subjects included in the present study were stratified by random sampling by age and sex from the CEEC project. At the time of saliva collection, demographic and lifestyle information was collected by trained investigators using face-to-face interviews. The collected information mainly included demographic data (age, gender, and education) and lifestyle habits (cigarette smoking status, alcohol drinking status, and oral health status). The information of oral health status included (1) whether any teeth were lost after age 20; (2) the number of teeth lost after age 20; (3) the frequency of tooth brushing. Alcohol drinking status information included: (1) whether they have a regular alcohol drinking habit (drink at least once a week for six months); (2) years of regular drinking habit; (3) the frequency of alcohol drinking; (4) the most common alcohol consumed and the average volume per time. Drinkers were defined as subjects who had been drinking at least once a week over six months (*N* = 54). In the drinker group, the average time of alcohol consumption habitat is 18.96 ± 14.84 years; the average frequency of drinking is 1.23 ± 2.51 times/week; the average volume per time is 3.49 ± 2.06 cups. The most common types of alcohol in this population are beer (75%) and baijiu (11%).

The collection of saliva has been described previously [18]. Briefly, unstimulated whole saliva samples were collected from participants during study enrollment. All participants were asked not to eat or drink for half an hour before providing samples. Five milliliters of saliva were collected. An equal volume of salivary lysate was added to the saliva to facilitate subsequent nucleic acid extraction. The salivary lysate included Tris-HCL (pH = 8.0), EDTA, sucrose, NaCl, and 10% SDS. The collected saliva samples were immediately placed on ice, and then stored at −80 °C within 4 h.

### 2.2. DNA Extraction and 16S rRNA Gene Sequencing

Saliva microbial DNA was extracted using the Powersoil DNA isolation kit (Qiagen, Duesseldorf, Hilden, Germany) according to the manufacturer’s instructions. Amplicon libraries were generated following the previous study [18]. Briefly, the V4 hypervariable regions of the 16S rRNA gene were amplified with 515F/806R primers [19] containing common adapter sequences and 12-bp barcodes with 20 cycles. Next, the Illumina flow cell adapters and dual indices (6bp) were added in a second amplification with 10 cycles of amplification. PCR products were purified using Agencourt AMPure XP (Beckman Coulter, Brea, CA, USA), and quantified using the Qubit HS kit (Invitrogen, Carlsbad, MA, USA). Finally, amplicon libraries were sequenced on the Illumina MiSeq PE-250bp platform (Illumina).

### 2.3. Bioinformatic Analysis of 16S rRNA Sequencing

16S rRNA gene amplicon sequences were processed and analyzed using the QIIME2 version 2019.4 [20]. Multiplexed libraries were demultiplexed based on the barcodes assigned to each sample. After they had been demultiplexed, paired-end read joining was performed with DADA2 [21]. Merged sequences were clustered into amplicon sequence variants (ASVs) and subsequently assigned to taxa with >99% identity by using the Greengenes database 13.8 [22] pre-defined taxonomy map of reference sequences. ASVs observed in fewer than three samples and a total abundance of less than five was excluded.

### 2.4. Statistical Analysis

We conducted the descriptive analysis to compare demographics between non-drinkers and drinkers. Chi-square tests and Student’s *t*-tests were used for categorical and continuous variables respectively. For the diversity analysis of 16S rRNA data, sample counts were rarefied to 10,000 sequences per sample. We assessed the alpha diversity of the saliva microbiota between non-drinkers and drinkers using Shannon’s diversity index and the inverse Simpson’s index [23]. We assessed the beta diversity using weighted UniFrac distance and Bray–Curtis distance matrices [24]. Permutational multivariate analysis of variance (PERMANOVA; adonis function, vegan package, R) was used to test differences in overall oral microbiome composition across non-drinkers and drinkers. Principal coordinate analysis (PCoA) was performed to obtain the first three principal coordinates of the above beta distance matrices and visualize complex multidimensional data. Linear regression with covariate adjustment was used to examine the difference of α-diversity indices and principal coordinates among drinking groups.

ASVs were classified into 74 genera and 42 species. We used the “DESeq” function [25] within the DESeq2 package in R to test for differentially abundant taxa between drinkers and non-drinkers at the genus and species level. This function models raw counts using a negative binomial distribution and adjusts internally for “size factors” which normalize for differences in sequencing depth between samples. Log_2_Fc (Log2 FoldChange) is the statistics of this analysis. A positive Log_2_Fc means that the abundance of taxa in drinkers is higher in comparison to non-drinkers; A negative Log_2_Fc means that the abundance of taxa in drinkers is lower in comparison to non-drinkers. Outlier counts were filtered out based on default Cooks distance cutoff threshold (automatic outlier filtering/replacement).

Functional prediction of 16S rRNA was conducted using Phylogenetic Investigation of Communities by Reconstruction of Unobserved States (PICRUSt) (http://galaxy.morganlangille.com/, accessed on 23 December 2021). PICRUSt can predict the Kyoto Encyclopedia of Genes and Genomes (KEGG) pathway functional profiles of microbial communities via 16S rRNA gene sequences [26]. The linear discriminant analysis effect size (LEfSe) method [27] was used to identify differentially abundant KEGG pathways. We used Spearman’s rank correlation to examine the associations between pathways and genera that were significantly associated with drinking status.

All statistical tests were two-sided, and the *p*-value < 0.05 was considered statistically significant. All statistical analyses were performed using R version 3.6.3.

## 3. Results

### 3.1. Characteristics of the Study Participants

We conducted an oral microbial study of 150 individuals to study the association between oral microbiota and alcohol drinking. The study flow chart was shown in Appendix A. Among 150 subjects, 64% (*N* = 96) were non-drinkers and 36% (*N* = 54) were drinkers. The demographic characteristics of non-drinkers and drinkers are shown in Table 1. Age and education level were equally comparable between the two groups. Sex is the factor significantly different between the two groups. The drinkers had higher percentages of males and were more likely to smoke, which is consistent with the previous perceptions [12]. Oral health status information, including whether teeth were lost after age 20, the number of lost teeth after age 20, and tooth brushing frequency were comparable between drinker and non-drinker group.

### 3.2. The Influence of Alcohol Drinking on Oral Microbial Diversity and Overall Composition

To investigate the effects of alcohol drinking on oral microbial diversity and overall composition, we evaluated α-diversity and β-diversity indices between non-drinkers and drinkers. Age, sex, and cigarette smoking were adjusted in the following analysis. The InvSimpson index, reflecting the species evenness, was significantly higher in drinkers than in non-drinkers (*p* = 0.039, Figure 1a). The Shannon diversity index, reflecting the richness and evenness of species, tended to be higher in drinkers with no statistical significance observed (Figure 1b). Results above showed that drinkers tend to harbor a more diverse oral microbiome than non-drinkers.

Next, we performed the PCoA based on weighted UniFrac distances and Bray–Curtis distance to determine whether the microbial composition differed according to alcohol drinking. We observed significant differences between non-drinkers and drinkers on weighted UniFrac distance (R^2^ = 0.0275, *p* < 0.001, Figure 2a) and Bray–Curtis distance (R^2^ = 0.0269, *p* < 0.001, Figure 2b) according to PERMANOVA analysis after controlling for covariates. We also calculated the difference of the first, second, and third coordinates of PCoA for non-drinkers and drinkers. Results showed that drinkers differed from non-drinkers in the first principal coordinate in PCoA in the weighted UniFrac distance (*p* = 0.0030, Figure 2c) and the Bray–Curtis distance (*p* = 0.0038, Figure 2d). These results showed that the oral microbiome community was significantly different between drinkers and non-drinkers.

### 3.3. The Influence of Alcohol Drinking on Oral Microbial Taxa

To further explore the specific bacteria associated with alcohol drinking, negative binomial generalized linear models were performed to compare the relative abundance of bacterial taxa between non-drinkers and drinkers at genus and species levels. Analysis was controlled covariates of age, sex, and smoking status. Taxa with significant abundance between the drinker and non-drinker groups were shown in Table 2. At genus level, two taxa were enriched in the drinkers as compared to non-drinkers, including *Prevotella* (log_2_ fold change [log_2_F_C_] = 0.47, *p* = 0.0033) and *Moryella* (log_2_F_C_ = 0.58, *p* = 0.042). Meanwhile, three taxa were depleted in drinkers, including *Lautropia* (log_2_F_C_ = −0.82, *p* = 0.0039), *Haemophilus* (log_2_F_C_ = −0.31, *p* = 0.0093) and *Porphyromonas* (log_2_F_C_ = −0.29, *p* = 0.036). At species level, *Prevotella melaninogenica* (log_2_F_C_ = 0.68, *p* = 0.0016) and *Prevotella tannerae* (log_2_F_C_ = 0.87, *p* = 0.0024) were enriched in the drinkers as compared to non-drinkers; *Haemophilus parainfluenzae* (log_2_F_C_ = −0.29, *p* = 0.021) were depleted in drinkers. We also compared the taxa abundance in phylum, class, order, family, genus and species levels. Three models were constructed with controlling different factors: model1 (none factor adjusted), model 2 (age, sex and smoking status adjusted) and model 3 (age, sex, smoking status and oral health status adjusted). After adjustment for oral health status, 16 of 20 the alcohol drinking related bacteria remained significant (shown in Appendix A).

### 3.4. Correlation between Oral Microbiota and Predictive Functional Pathways

We performed PICRUSt analysis to investigate the functional difference of the oral microbiota between non-drinkers and drinkers. Of 328 KEGG pathways identified, 21 non-human-gene pathways differed in abundance between non-drinkers and drinkers (*p* < 0.05 in LEfSe analysis, Appendix A). Among these pathways, 16 belonged to metabolism, others belonged to genetic information processing (*n* = 3), environmental information processing (*n* = 1), and cellular processes (*n* = 1). Next, we analyzed the correlation between differentially abundant pathways and genera to explore whether the bacteria altered by alcohol consumption were necessarily related to these pathways (Figure 3). We observed that the bacteria significantly enriched in drinkers had similar functions but had a distinct difference in function from those decreased in drinkers. In metabolism pathways, genera enriched in drinkers were positively associated with the metabolism of cofactors and vitamins, glycan biosynthesis and metabolism. Additionally, genera enriched in drinkers were negatively associated with nucleotide metabolism, metabolism of other amino acids, lipid metabolism, enzyme families, and amino acid metabolism. In carbohydrate metabolism, pathways including galactose, fructose and mannose metabolism pathways were enriched in drinkers and positively associated with genera enriched in drinkers, while the pyruvate metabolism pathway was decreased in drinkers and negatively associated with genera enriched in drinkers.

## 4. Discussion

Dysbiosis of the oral microbiome is linked to multiple diseases. Identifying the environmental factors affecting oral microbiota will contribute to our understanding of oral microbiota and its role in pathogenesis. Several previous studies have linked alcohol drinking to oral dysbiosis, with effects on the diversity and composition of oral microbiota [12,13,14]. The oral microbial community is shaped by lifestyle, diet, living conditions, and geography [16,28], though whether the previously found association of oral microbiota with alcohol consumption could be validated in different populations is unclear. Studies are still needed to provide more evidence from a diverse population. In the present study, we carried out an oral microbial study of 150 individuals to characterize microbial alterations in alcohol drinkers and non-drinkers. Our results showed clear differences in the community structure and composition, as well as the functional profiles of the oral microbiota between non-drinkers and drinkers.

Compared with non-drinkers, drinkers had a significantly higher alpha diversity and significantly different structures. These results suggest that alcohol drinking might influence the stable balance of oral microbiota, consistent with the large population-based study published by Fan et al. [12]. Previous studies showed that alcohol intake was associated with the risk of periodontal diseases and caries [29,30]. Jiyoung Ahn et al. also found that the diversity of oral microbiota differed between heavy drinkers and non-drinkers, with drinkers having higher richness and evenness [12]. Evidence from a longitudinal study also showed that microbial diversity of patients with alcohol use disorder was decreased during the period of abstinence, coupled with the decreasing abundance of periodontal disease-associated genera [31]. Saliva plays a major role in maintaining the composition and activity of the oral microbiota [32]. Continuous alcohol consumption alters the oral ecosystem, including reducing saliva production, and disrupting tooth enamel, resulting in cavities and periodontal inflammation [8,9]. Dysbiosis can occur rapidly if the flow of saliva is perturbed [32].

We further characterized the microbiota features related to alcohol drinking at the genus level and species level. At the genus level, we observed *Prevotella* and *Moryella* were enriched in the drinkers as compared to non-drinkers, while genus *Lautropia*, *Haemophilus*, and *Porphyromonas* were depleted in drinkers. The finding that a higher abundance of oral *Prevotella* in alcohol drinking was consistent with previous studies [12,14]. The enrichment of *Prevotella* has also been observed both in the gut of humans and mice after alcohol consumption [33,34]. These results indicated that the effect of alcohol intake was extensive in causing perturbations both to the oral and gut microbiome. Besides, the previous 16S rRNA gene sequences on alcohol drinking and oral microbiota at higher resolution have been less thoroughly investigated. At the species level, we observed *Prevotella melaninogenica* and *Prevotella tannerae* were enriched in the drinkers as compared to non-drinkers after controlling for age, sex, and smoking status, while species *Haemophilus parainfluenzae* was depleted in drinkers, observations which were first identified in our study.

Several studies have found close relationships between specific genera enriched in drinkers and a variety of diseases. *Prevotella*, a dominant genus in the family *Prevotellaceae*, has been implicated in multiple diseases including inflammatory autoimmune disease [35], opportunistic infections [36,37,38,39], bacterial vaginosis [40], and oral cancer [41,42]. *Prevotella* spp. including *P. melaninogenica* are the species most frequently associated with infections in humans [43]. The mechanism for *Prevotella* participating in human disease has been thoroughly investigated. Animal and human studies have indicated that *Prevotella* promotes chronic inflammation by activating Toll-like receptor 2 predominantly [44]. *Moryella* is a genus belonging to the family *Lachnospiraceae*, which has been reported to be involved in bacteremia [45], and gynecologic cancer [46]. While for taxa depleted in drinkers, *Haemophilus* spp. is a ubiquitous oral commensal that has been found in lower abundance in OSCC patients compared to healthy controls [47]. Furthermore, oral commensal *Haemophilus parainfluenzae* was observed anticancer properties with in vitro study [48]. Thus, it is possible that alcohol drinking causes perturbations to the oral microbiome by contributing to an overgrowth of pathogenic bacteria and a decrease of health-related bacteria.

We explored microbiota function based on inferred metagenomes to profile the functional alterations due to alcohol drinking and to identify whether the oral microbes associated with alcohol drinking had different biological functions. We found that the pathways with significant differences between non-drinkers and drinkers mainly belong to the metabolism category. Among them, oxygen-independent pathways (galactose, fructose and mannose metabolism pathways) [49] were enriched in drinkers. Pyruvate metabolism, an aerobic metabolism pathway [50] was decreased in drinkers compared with non-drinkers. This suggested that alcohol drinking may create an oxygen-starved environment in the oral cavity through its influence on oral microbiota. The oxygen-starved environment is characterized by extremely low oxygen concentrations and is considered to be a perfect niche for the growth of pathogenic bacteria [51], which has been observed in dental caries [51] and nearly all solid tumors [52]. Additionally, we found that bacteria that were significantly enriched in drinkers had opposite functions to those bacteria depleted in the drinkers. These findings suggested that alcohol drinking may affect oral health by altering the microbiota and their metabolic functions.

There are some limitations in our study. First, in this study, although we obtained questionnaire-based oral health status information, detailed information on oral health examinations is lacking. Oral health and diseases strongly affect the oral microbiome [53]. Studies with professional oral examination information could provide more accurate evidence about the association between the oral microbiome and alcohol drinking. Second, the sample size in this study is relatively small, which makes it difficult to determine the low abundance taxa associated with alcohol consumption. Studies with larger sample size are needed to validate our findings and explore more associated taxa. Third, considering the sample size in this study, we did not conduct further stratified analysis on factors such as frequency of drinking, type of drinking, etc. The dose and type of alcohol consumption may influence the effect of alcohol consumption on the microbiota [54]. A comparison of characteristics of alcohol consumption and oral microbiota might improve the robustness of the findings. In this study population, individuals’ systematic disease information is lack. Considering that alcohol consumption could also impact general health, the missing information on common diseases could confound the results.

To the best of our knowledge, our study included the largest number of samples in the Chinese population to demonstrate the effects of alcohol drinking on the oral microbiota composition. Our results not only confirmed the presence of oral microbiota dysbiosis in drinkers but also further revealed specific bacteria that were significantly affected by alcohol drinking, as well as the potential biological function of oral microbiota. Future studies with a larger sample size and sufficient metagenomic data are still needed to investigate the impact of alcohol drinking on the oral microbiota taxonomic and functional features. Furthermore, attention should be paid to the possible mechanisms of the alcohol-induced oral microbiota changes in disease.

## 5. Conclusions

In summary, in this oral microbiota study, we observed that alcohol drinking led to altered oral microbiota community structure and higher diversity. The abundances of specific oral taxa were associated with drinking status. Our study suggested that alcohol drinking may affect health by altering certain metabolic pathways of oral microbiota.

## Figures and Tables

**Figure 1 ijerph-19-05729-f001:**
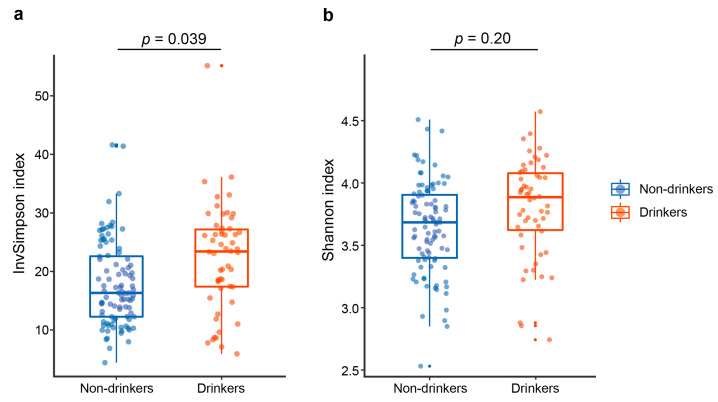
Alpha diversity estimates of the oral microbial community. Comparison of (**a**) InvSimpson index and (**b**) Shannon index in the oral microbiota between non-drinkers and drinkers. *p* values were calculated by the linear regression model.

**Figure 2 ijerph-19-05729-f002:**
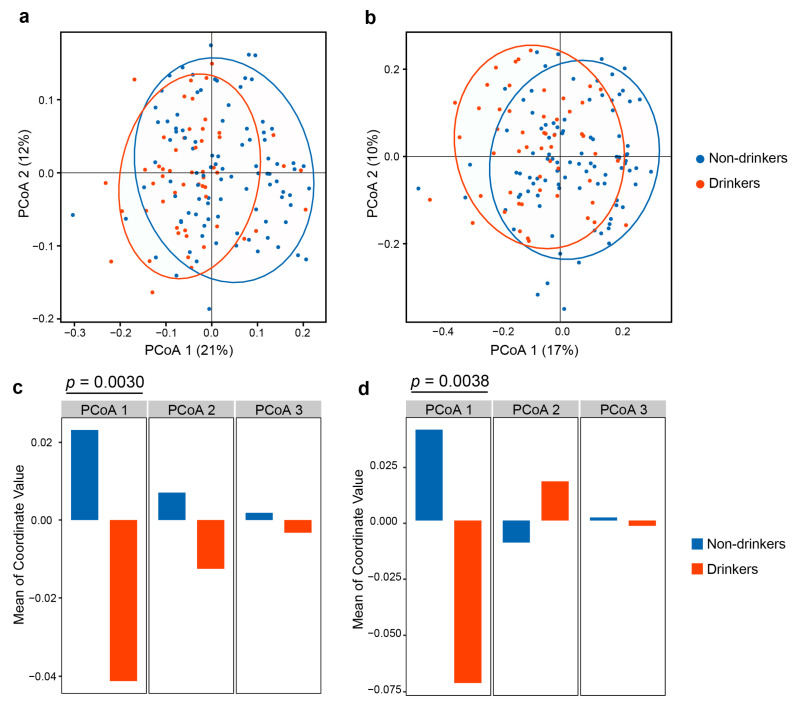
Beta diversity estimates of the oral microbial community. a, b PCoA based on (**a**) weighted UniFrac distance matrix and (**b**) Bray–Curtis distance matrix of the oral microbial communities between non-drinkers and drinkers. (**c**,**d**) Bar plots showing the means of the first, second, and third coordinates of PCoA for non-drinkers and drinkers using (**c**) weighted UniFrac and (**d**) Bray–Curtis distance matrix. *p* values were calculated by the linear regression model with controlling age, sex and smoking status.

**Figure 3 ijerph-19-05729-f003:**
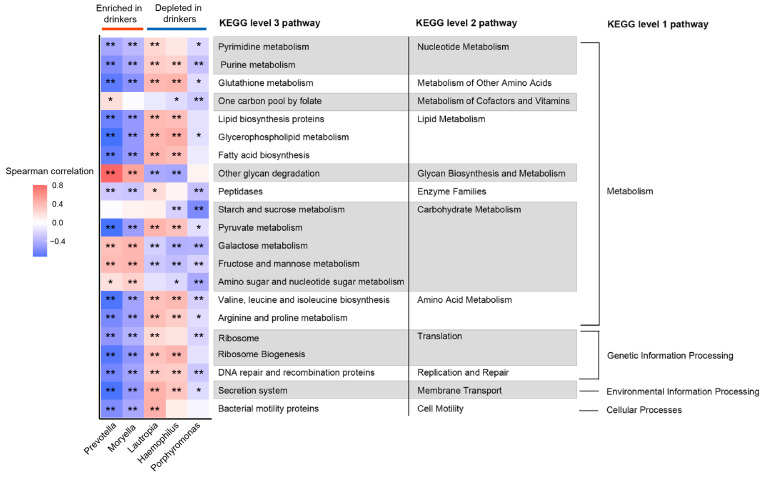
The genera associated with drinking status are related to several gene functional pathways. Spearman’s correlation coefficients were estimated for each pairwise comparison of genus counts and KEGG pathway counts. The strength of the color depicts Spearman’s correlation coefficients (negative score, blue; positive score, red). * *p* < 0.05, ** *p* < 0.01.

**Table 1 ijerph-19-05729-t001:** Demographic characteristics among non-drinkers and drinkers.

	Non-Drinkers (*N* = 96)	Drinkers (*N* = 54)	*p*-Value
Age, mean (s.d.)	45.63 (9.73)	49.24 (11.04)	0.05 ^1^
Sex, *n* (%)			<0.01 ^2^
Male	58 (60.42)	48 (88.89)	
Female	38 (39.58)	6 (11.11)	
Education, *n* (%)			0.68 ^2^
<High school	77 (80.21)	41 (75.93)	
≥High school	19 (19.79)	13 (24.07)	
Smoking status, *n* (%)			<0.001 ^2^
Non-current	61 (63.54)	15 (27.78)	
Current	35 (36.46)	39 (72.22)	
Teeth loss after age 20, *n* (%)			0.13 ^2^
Yes	45 (46.88)	33 (61.11)	
No	51 (53.12)	21 (38.89)	
The number of teeth lost after age 20, mean(s.d.)	3.33 (7.08)	3.54 (6.15)	0.85 ^1^
Tooth brushing frequency, *n* (%)			0.84 ^2^
≤1 time a day	61 (63.54)	36 (66.67)	
≥2 times a day	35 (36.46)	18 (33.33)	

^1^ *p*-value was based on Welch Two Sample *t*-test. ^2^ *p*-value was based on Pearson’s Chi-squared test.

**Table 2 ijerph-19-05729-t002:** The differential bacterial taxa between non-drinkers and drinkers at genus and species levels.

Taxonomy	Mean Count ^1^		log_2_Fc (95% CI) ^3^	*p*-Value ^2^
	Non-Drinkers (*N* = 96)	Drinkers (*N* = 54)		
Genus level				
*Prevotella*	1255.23	1810.96	0.47 (0.16, 0.78)	0.0033
*Moryella*	17.05	32.07	0.58 (0.02, 1.14)	0.042
*Lautropia*	107.90	65.59	−0.82 (−1.37, −0.26)	0.0039
*Haemophilus*	780.65	598.50	−0.31 (−0.54, −0.08)	0.0093
*Porphyromonas*	557.63	488.76	−0.29 (−0.56, −0.02)	0.036
Species level				
*Prevotella melaninogenica*	447.50	805.52	0.68 (0.26, 1.10)	0.0016
*Prevotella tannerae*	53.61	109.00	0.87 (0.31, 1.43)	0.0024
*Haemophilus parainfluenzae*	726.85	567.28	−0.29 (−0.54, −0.04)	0.021

^1^ Sequence read counts were rarefied to 10,000 sequences per sample.^2^ *p*-value was calculated by DESeq function, adjusted for age, sex, and smoking status. ^3^ Log_2_Fc > 0: the abundance of taxa in drinkers is higher in comparison to non-drinkers; Log_2_Fc < 0: the abundance of taxa in drinkers is lower in comparison to non-drinkers.

## Data Availability

The data and materials used in this study are available from the corresponding author on reasonable request.

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
