# Peer review of "The Effects of Alcohol Drinking on Oral Microbiota in the Chinese Population"

_ijerph, 2022, doi:10.3390/ijerph19095729_

Round 1

Reviewer 1 Report

This manuscript is focused on the influence of drinking on oral microbiota, but there was the logical problem of contents, some mistakes and concepts must be clarified and revised. This paper should be totally reconstructed again.

Author Response

Dear Reviewer,

Thank you for your comments. The coverletter can be found at the attachment.

Reviewer 2 Report

In this study, researchers analyze the effects of alcohol consumption on the diversity and composition of the oral microbiome. They found that alcohol significantly influenced the diversity of the microbiome and that several distinct metabolic pathways in microbes may influence abundance in drinkers. This study and similar studies are contributing to our growing understanding of how behavior/diet can influence health by shaping the human microbiota. The study was well designed, conducted, and analyzed and the data is clearly and accurately presented. I believe it is suitable for publication. I have listed some small grammatical errors and comments below:

Line 51- have direct cytotoxic effects

Line 55- oral microbiota communities

Line 57- enriched

58- conducted not investigated

151- biological sex not gender. Sex, age, and smoking are well known to influence the human microbiota. Find and replace all instances of gender to sex.

 you can remove “potential covariates” these are known covariates

199- alcohol consumption

215- Disymbiosis of the oral microbiome is linked….

This study makes several references to the Chinese population, which is fine to recognize the demographics and composition of your study group, but the way this phrase is used it makes it seem like results in other studies might not be applicable to the Chinese population and vice versa. And that might indeed be possible if the composition of the oral microbiome is influenced by geography i.e. “however, the effect of alcohol drinking on the composition of the oral microbiota in Chinese people remains unclear” Are you suggesting that the studies you reference in your introduction do not apply to the Chinese population? Should the results of this study be understood to only impact Chinese people? If you think the results of these studies should not be universalized across countries, then you need to explain this in the discussion or remove references to the Chinese population.

Author Response

(The authors gave the same response as above.)

Round 2

Reviewer 1 Report

Some questions mentioned before didn't really revise all, but the problem mentioned is an extremely important effect on the real result, so the material and method and the experiment of the manuscript must be reconstructed and do it again.
